# A Perspective on Organoids for Virology Research

**DOI:** 10.3390/v12111341

**Published:** 2020-11-23

**Authors:** Adithya Sridhar, Salvatore Simmini, Carla M. S. Ribeiro, Caroline Tapparel, Melvin M. Evers, Dasja Pajkrt, Katja Wolthers

**Affiliations:** 1OrganoVIR Labs, Department of Medical Microbiology, Amsterdam UMC, Location Academic Medical Center, University of Amsterdam, 1100 AZ Amsterdam, The Netherlands; a.sridhar@amsterdamumc.nl (A.S.); d.pajkrt@amsterdamumc.nl (D.P.); 2Department of Pediatric Infectious Diseases, Emma Children’s Hospital, Amsterdam UMC, Location Academic Medical Center, University of Amsterdam, 1100 AZ Amsterdam, The Netherlands; 3Gastrointestinal Biology Group, STEMCELL Technologies UK Ltd., Cambridge CB28 9TL, UK; salvatore.simmini@stemcell.com; 4Department of Experimental Immunology, Amsterdam Institute for Infection and Immunity, Amsterdam UMC, University of Amsterdam, 1100 AZ Amsterdam, The Netherlands; c.m.ribeiro@amsterdamumc.nl; 5Department of Microbiology and Molecular Medicine, Faculty of Medicine, University of Geneva, 1211 Geneva, Switzerland; Caroline.Tapparel@unige.ch; 6Division of Infectious Diseases, Geneva University Hospital, 1205 Geneva, Switzerland; 7Department of Research and Development, uniQure Biopharma B.V., 1105 BE Amsterdam, The Netherlands; m.evers@uniqure.com

**Keywords:** human organoids, virology, standardization

## Abstract

Animal models and cell lines are invaluable for virology research and host–pathogen interaction studies. However, it is increasingly evident that these models are not sufficient to fully understand human viral diseases. With the advent of three-dimensional organotypic cultures, it is now possible to study viral infections in the human context. This perspective explores the potential of these organotypic cultures, also known as organoids, for virology research, antiviral testing, and shaping the virology landscape.

Because viruses are obligate intracellular parasites, model systems comprising various cellular components are necessary for elucidating their biology. Historically, animal models and immortalized (cancerous) cell lines enabled virology research and host–pathogen interaction studies. Although these model systems immensely contributed to expand our knowledge of virology, the limitations of these models are clear [1]. Animal models do not adequately reproduce human disease pathology, and in some instances, viral pathogens have a unique human tropism that cannot be replicated in a non-natural host. Similarly, cell lines display intrinsic alterations in major signaling pathways, and thus do not recapitulate the homeostatic functions of a normal cell. Cell lines also lack the structural and functional complexity of an organ. These deficiencies often lead to a virus adapting to a cell line after a single passage. This caveat calls for novel validated models that can recapitulate human physiology and are predictive of human disease. Human organoids and organotypic cultures could address this unmet need for human model systems in virology research and antiviral testing. Therefore, in this perspective, we (1) provide an introduction to organoid technology, (2) explore how it could shape the virology landscape, (3) comment on challenges for widespread implementation, and (4) speculate on the future of this technology in the context of viral research. 

## 1. Organoid Technology

Organoids are self-organized 3D cultures derived from stem cells. They are a miniature and simplified version of an organ that recapitulates the genetic phenotype, organization, and (limited) functionality of the organ. They consist of organ-specific cell types with spatially restricted lineage commitment. Organoids can be derived from induced pluripotent or multipotent tissue stem cells that can be embryonic, fetal, or adult in origin. Organoids have been established for several different organ systems: intestine, stomach, lung, liver, thyroid, kidney, blood vessel, brain, prostate, pancreas, and ovaries. For the sake of simplicity, we include human airway epithelial (HAE) cultures under this umbrella, as they are widely used in virology. HAE cultures are derived by culturing primary airway epithelial cells on permeable inserts and differentiating them in an air–liquid interface [2]. HAE cultures, by definition, are not organoids, but these organotypic cultures recapitulate the cellular heterogeneity and function of the airway, offering the same advantages as those of organoids. The main benefit of using these stem-cell-derived organoid and HAE models is their ability to mimic pathologies at the organ level. Furthermore, the use of human stem cells allows for recapitulating human-specific features that make them relevant for translational studies. For a complete overview of organoid systems, and their potential and applications, several excellent reviews cover this topic in depth [3,4,5].

## 2. Organoids Broaden the Scope for Virology 

Specific viruses cause diverse diseases in different animal models, and infection follows distinct routes within the host. For this reason, it is important to employ the most natural system to study the effects of viral infections. In order to increase the translatability of results from an in vitro or ex vivo model to an in vivo situation, it is imperative to rely on results from human-based model technology. Virus-infected human organoids can provide a more accurate image of what host factors are essential for the establishment of viral infection in humans, and improve the study of effects between donors with varying age, sex, or genetic make-up. The identification of such factors is essential in understanding why some individuals experience only mild disease after a specific viral infection, while others fall severely ill or even die. If the response to a viral infection can be predicted for individuals or groups, this leverages more personalized medicine, such as who to hospitalize or treat. 

### 2.1. Organoids Enable Culturing the Unculturable

Most virology research groups use immortalized cell lines to grow viruses. For example, HeLa cells are widely used to study rhinoviruses that cause respiratory disease, even though this cell line is derived from human cervical tissue. Therefore, to efficiently grow in these artificial culture models, viruses often need to adapt. For some viruses, e.g., the norovirus, this adaption is not possible, and despite intensive trials in multiple cell lines, they remain unculturable. Stem-cell-derived human intestinal organoids have allowed for the culture of these unculturable or hardly culturable enteric viruses, such as norovirus, although it is still limited to one round of infection [6]. Similarly, respiratory viruses that are unculturable in cell lines were successfully isolated from clinical specimens using HAE cultures. These models were used to grow and characterize human coronavirus HKU1, human bocavirus, and human rhinovirus C [7,8,9]. The permissiveness of these models for the rapid propagation of viruses directly from clinical isolates makes them an interesting and relevant platform for the characterization of unknown viruses [10]. 

### 2.2. Organoids Recapitulate the Natural Virus Host Environment

Viruses can be directly amplified in organoid models from clinical isolates without the need to mutate or adapt. In contrast to laboratory-adapted viral strains or American Type Culture Collection (ATCC) strains grown in immortalized cell lines, viruses from infected human materials (such as feces, blood, or nasopharyngeal swabs), grown only in organoids or HAE, more closely retain their original characteristics and infectivity profiles [11,12,13]. Consequently, findings about their tropism, receptor usage, and innate immunity induction are highly relevant and exclude any bias linked to laboratory cell adaptation. 

#### 2.2.1. Organoids Allow for Identification of Entry Host Receptors and Pathways

The expression of surface molecules may be completely different in immortalized cells or animal models compared to human host tissue, and results regarding receptor usage can be misleading and result from lab adaptation [14]. An example of this is the acquired dependency on heparan sulfate binding through cell-culture adaptation [15]. Only a few immortalized cell lines are polarized, and do not always recapitulate the receptor distribution observed on polarized organoid cultures or in vivo situations. Here, organoids allow for dissecting the polarized entry of viruses and their ability to cross epithelial barriers. The pathogenesis of the measles virus (MV) is an example where this has long been misunderstood. For years, it was thought on the basis of in vitro findings that the infection started with entry at the apical side of respiratory epithelia. The first discovered MV cellular receptor was CD46, a cell-surface molecule ubiquitously expressed by most nucleated cells, but subsequent studies showed that the CD46-mediated entry is limited to vaccine- and laboratory-adapted MV strains [16]. Recently, the use of complementary in vitro, ex vivo, and in vivo approaches highlighted that respiratory epithelial cells cannot be the initial targets of MV because nectin-4, the true receptor of wild-type MV strains, is only expressed at the basolateral side of epithelial cells, and is thus not accessible upon entry from the lumen side of respiratory tissue [17]. Epithelial cells are, therefore, infected later from the basolateral side and, combined with epithelial damage, lead to the release of particles in the respiratory tract and MV transmission [18]. 

#### 2.2.2. Infected Organoids Resemble Clinical Observation In Vivo

Brain organoids were extensively used to confirm the neurotropism of the Zika virus and its preferential infection of neural progenitor cells [19]. Zika virus infections gained global attention because children born with a Zika virus infection showed severe neurological complications such as microcephaly [20]. This phenomenon was illustrated with the use of human-brain organoids derived from induced pluripotent stem cells (iPSC). The Zika virus infection reduced the size of human brain organoids, mimicking the Zika-virus-induced microcephaly in affected children. Besides increased insight into the mechanism of Zika virus pathogenesis in humans, efforts are now underway to investigate routes of infection, neurotropism, neurovirulence, and related effects in iPSC-derived human-brain organoids for herpes-simplex-virus and human-cytomegalovirus infections [21,22]. Similar to Zika virus infection, neonatal herpes-simplex-virus and congenital-cytomegalovirus infections can elicit major neurological deficits such as microcephaly. With the use of human-organoid technology, these viral infections can be more effectively studied, potentially leading to new effective therapeutic interventions [23].

### 2.3. Organoids Provide New Insights 

The use of intestinal organoids highlighted the critical role of an enterovirus upstream open reading frame (uORF) present in the 5′UTR genomic region. Many enteroviruses present a small open reading frame (ORF) upstream of the main polyprotein ORF. Early studies performed in cell lines concluded that this uORF is not used for translation initiation [24]. Recently, Lulla and colleagues could show that, although dispensable in cell lines, the small protein encoded by this uORF plays an essential role for virus release in human intestinal organoids and that viruses lacking this uORF are attenuated in this model [25]. Further, the specific cellular tropism and response to infection can also be dissected in tissue culture models. Recent infection studies in intestinal organoids showed that different enteroviruses infect different cell types and induce a cell-type-specific antiviral response [26,27].

Another interesting advantage of using organoid models is the ability to study the interactions between codetected pathogens, which is not always possible in cell lines because viruses might be culturable on only two different cell lines. For example, respiratory syncytial virus grows better on A549, influenza virus on MDCK, and rhinovirus on HeLa cells; thus, it is difficult to find a cell line that is equally suited for the growth of these three pathogens [28]. These organoid models can also be used to assess the effect of host conditions such as age and comorbidities on viral infections [29,30]. Respiratory viruses exacerbate asthma or other respiratory comorbidities such as cystic fibrosis or chronic obstructive pulmonary disease. HAE infection derived from healthy and asthmatic donors with rhinovirus highlighted a different airway epithelial structure and inflammatory signaling in asthmatic patients [29].

### 2.4. Organoids Are Valuable Tools in the Severe Acute Respiratory Syndrome Coronavirus 2 Pandemic

Organoids were widely applied to study severe acute respiratory syndrome coronavirus 2 (SARS-CoV-2) recapitulating clinical disease and they provide novel insights highlighting their utility for virology research [31]. HAE cultures were used as a model of the primary replication site, demonstrating efficient replication in this airway through the infection of ciliated cells [32]. A plaquelike cytopathic effect (CPE) was observed with loss of epithelial integrity, shrinking cilium, cell fusion, and apoptosis [33]. Therapeutic evaluation on organoid models showed remdesivir and remdesivir–diltiazem to be effective against SARS-CoV-2 infection [34].

In addition to enabling indepth studies of clinical observations of lung epithelial infection, the potential to infect secondary tissue was also observed using organoid models. Lamers et al. showed that SARS-CoV-2 productively infects the human gut epithelium, targeting differentiated enterocytes [35]. Similarly, Stanifer et al. showed productive infection of human intestinal epithelial cells with a robust intrinsic immune response, and demonstrated the critical role of Type III interferon in controlling infection [36]. These organoid studies provided novel insights, as the SARS-CoV-2 genome is detectable in feces even after the virus is not detectable in oropharyngeal swabs, raising concerns of intestinal infection and potential fecal transmission [37]. 

Another area of investigation using organoids has been determining the neuroinvasive potential of SARS-CoV-2 [38]. Epidemiological studies showed that SARS-CoV-2 infection can present with neurological complications such as headaches, ischemic stroke, and encephalitis along with cranial- nerve-related complications such as anosmia, hyposmia, and ageusia. This has led researchers to employ brain organoids for assessing if SARS-CoV-2 has neuroinvasive potential. Zhang and colleagues showed that SARS-CoV-2, but not SARS-CoV, can infect human neural progenitors [39]. Furthermore, they demonstrated that SARS-CoV-2 can productively infect human brain organoids, targeting cortical neurons. Similarly, Ramani et al. also showed infection of brain organoids through neurons, resulting in neuronal death [40]. Moreover, recent data from Pellegrini et al. using choroid plexus organoids demonstrated potential viral tropism for choroid plexus epithelial cells, resulting in damage to the epithelium [41]. Damage to this barrier could present a potential route of entry for the virus into the cerebrospinal fluid and the brain. Taken together, these studies illustrated the promise of organoids for studying emerging viral infections, and elucidated their full disease potential against clinical observations. 

## 3. Technical Challenges

While the previous section highlights the potential that organoids hold for virology, they are still a nascent technology with broader and virology-specific issues that need to be addressed as summarized below.

### 3.1. Technical Challenges Related to Virology

Organoids, as they were originally established, are closed round structures embedded in Matrigel^®^. These round structures are difficult to infect with viruses: receptors needed for infection are mostly located inside the organoid. As with HAE cultures, round gut organoids can be transformed to an open organoid model where cells grow on a Transwell^®^, so that they can be reached from the upper and lower sides to establish infections [6,26,42,43]. This model system is ideal for infectious-disease studies and to test antimicrobial drugs. 

Another important consideration is readouts used for analysis after infection. For instance, virus cultures in primary models often do not result in a CPE. This is likely due to the release of viral particles in a nonlytic manner, as seen in the case of enterovirus A71. Huang and colleagues demonstrated that enterovirus A71 infection of human intestinal organoids resulted in viral release through exosomes rather than lytic processes observed when using a classical RD cell line [44]. Moreover, the production of infectious viral particles after infection is assessed by performing back titration or plaque assays using cell lines. However, as described earlier, distinct behavior in primary culture versus cell lines raises concerns about using titer data from cell lines as a measure for evaluating the production of infectious viral particles in primary culture. More suitable methods need to be developed for directly assessing this in primary cultures. 

### 3.2. Standardization

Organoids are generated from primary cells, and culture conditions to maintain appropriate long-term cell composition and function are based on the knowledge of their in vivo requirements. For example, intestinal organoid culture systems typically require the activation of Wnt and EGF pathways, and the inhibition of BMP pathways. However there are variations of culture conditions for modulating these pathways, resulting in shifts in the cell composition of these organoid systems [3]. A striking example is illustrated by differences in cellular composition observed in human intestinal organoids derived in different medium formulations. The first described organoids exhibited a high proportion of stem cells shown by high expression levels of Wnt-related genes such as LGR5, ASCL2, AXIN2, and OLFM4, and the inhibition of goblet and enterocyte differentiation [45]. The subsequent withdrawal of Wnt from this medium triggered the generation of differentiated intestinal cell types. Fujii et al. developed culture media that gave rise to organoids exhibiting increased multilineage differentiation while sustaining high levels of self-renewal, as demonstrated by the expression of LYZ, CHGA, and MUC2 with LGR5 in a single culture condition [46].

Another factor that could introduce variability in organoid cultures is related to the starting material. Varying performance is often seen in different batches of conditioned media because of the difficulty to precisely control which factors are secreted into the medium. Further, Matrigel^®^ is not completely defined and may lead to some variability. In addition, the costs of 3D culture systems tend to be higher than those of standard 2D culture systems due to their complexity. Many laboratories find alternative approaches that influence the selection of media-component suppliers or the type of extracellular matrix, and develop internal production of serum-based conditioned media.

Reducing variability is hindered by a lack of widely accepted criteria and guidelines for assessing the quality and characteristics of organoid cultures that are generated using different reagents. During the past decade, protocols using different reagents and culture conditions were used for deriving and culturing organoids from a range of organs [47]. As discussed above, these limitations of current culture conditions in some of these systems can create bias toward specific cell types within organs. Thus, it is important to continue to improve and standardize culture media to make organoids more closely resemble their respective tissues in vivo. In fact, donor-source variability and organoid cultures derived from these tissue types are intrinsically heterogeneous; the use of different media can increase the variability and limit the reproducibility of outcomes measured in downstream applications. This lack of standardization is a major bottleneck and limits the comparison of results based on organoids generated through different protocols and research groups. A joint effort by the academic and industrial sectors is needed to optimize media formulations and enable the production of highly reproducible organoids with the goal of reducing experimental variability, increasing result accuracy, and reducing the cost of current commercial solutions.

### 3.3. High Throughput

Apart from the heterogeneous nature of currently developed organoids, the next challenge is the development of high-throughput platforms that support organoid technology. For the industrial and commercial use of organoid technology for translational applications such as antiviral screening, organoid technology needs to be scaled up with standardized quality control while maintaining the complex nature of the organoids. This quandary is one of the major limitations for widespread implementation of organoid technology in a commercial setting. For the adequate implementation of organoid models, validated cell-culture protocols are critically needed that demonstrate organoid cultures in high-density microplates (96-well, 384-well, or higher) with high quality-controlled standard performance to facilitate high-throughput screening (HTS). Validated HTS standards would facilitate reproducibility and reduce associated costs. The HTS methodology would be essential to commercially increasing the current use of organoids and facilitating personalized medicine.

## 4. Future Perspectives

### 4.1. Increasing Complexity 

At present, organoids are mainly single-organ systems representing the epithelium. For instance, gut organoids comprise the gut epithelial layer and lack the mesenchymal or immune-cell elements present in the gut mucosa [48]. The organoid community is undertaking efforts to address this issue through the establishment of cocultures of epithelial organoid systems with other organ- specific components. For instance, the cocultures of immune cells (such as macrophages and T cells) are possible with organotypic cultures such as intestinal organoids. This expedites studies on the role of the mucosal compartment and epithelial-immune cell communication in antiviral immunity [49,50]. Furthermore, the complexity of these in vitro model systems could be further improved by interconnecting multiple organ systems. Stem-cell-derived organoid models implemented in an organ-on-chip system enable the modeling of multiorgan complexity such as the gut–brain axis, allowing for the study of disease progression from primary to secondary infection sites. 

### 4.2. Functionality Measurement 

One of the advantages of animal models (in viral research) over human organoids is the availability of functionality readouts in animal systems. For example, in a monkey model, intravenous EV-A71 resulted in direct neurological signs such as tremor, ataxia, and brain edema [51]. Similarly, in mice models, poliovirus causes poliomyelitis with paralysis resembling paralytic disease in humans [52]. These functional observations are not yet possible in human organoid models. However, progress was made over the last couple of years. For example, in human brain organoid technology, Quadrato and Giandomenico demonstrated that the functional sensory input and motor output of organoids can be measured [53,54]. This indicates that human cerebral organoids could be useful in studying neural connectivity and other functional modalities of the brain. 

### 4.3. Replacing the Golden Standard

Currently, 500–1000 animals are needed for optimizing a new antiviral compound for clinical trials. However, 95% of drugs that proceed to the clinical-trial stage do not make it to the market despite promising results in animal models. On the basis of the resemblance to the in vivo situation, organoid models show potential superiority to animal models in predicting efficacy and toxicity of an antiviral compound and therefore reducing costs in the development of novel antiviral therapies. In the pharmaceutical industry, selecting and testing lead compounds to assess their potency and safety typically involves the use of animal models to show in vivo proof of concept and provide evidence for clinical benefit, which is time-consuming and resource-intensive. For some indications, animal models might be unavailable or not translatable due to a lack of human components that are required for their mechanism of action. For systemic indications, disease-relevant, patient-derived cellular models very often exist. However, for central or peripheral nervous system (CNS/PNS) disorders, relevant in vitro models derived from patients hardly exist due to the inaccessibility of nervous tissue. To overcome these limitations, patient stem-cell-derived brain organoids provide an important translational bridge: a reliable model for CNS/PNS indications in the human context, thereby reducing the number of animal studies required for both efficacy and mechanism-of-action safety studies, decreasing development time into the clinic, and increasing the success rate due to the better-translatable in vitro platform.

## 5. Conclusions

Over the past decade, organoids showed promise as a human model for studying human viral diseases. However, as a nascent technology, organoids still have room for advancement and standardization. As the complexity of these model systems increases with cocultures and organ-on-chip systems, new opportunities and challenges for the field arise, and the virology landscape benefits.

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
