# Peer review of "A Perspective on Organoids for Virology Research"

_viruses, 2020, doi:10.3390/v12111341_

Round 1

Reviewer 1 Report

The review-manuscript by Sridhar et al addresses a very interesting aspect of virus research, the use of organoids to determine viral function and interactions with the tissue microenvironment including the immune system. This is a relatively novel field in virology and is likely to play a very significant role in future virus research both in regard to basic molecular understanding of virus-host interactions and for clinical translation of anti-virals and oncolytic viruses. However, the manuscript does not describe in depth the various models that have been used so far, and specific findings are only briefly mentioned and only for a few viruses. My suggestion would be for the authors to add more specific data and illustrate models with figures. This could be a great and important review for all virologists if improved. 

Some suggestions for improvements include:

  • Illustrations of the various organoid and organotypic models that have been successfully applied in virology
  • Clear description of which cells were used and whether co-cultured with other cell types or grown alone or as a tissue specimen
  • A table listing the reported cultures with references and advantages - disadvantages  
  • In subsection 1 there is an example of HAE organotypic without essential details and no examples of organoids
  • Some examples are given later on in the text for MV and Zika viruses. It would be nice to have more information on these studies and add a few more results with other viruses. 
  • Subsection 3.3 talks about High-throughput screening but it is unclear what the authors want to say. Is this to screen for various viruses or anti-viral drugs or something else? Please give more details.

Author Response

We thank the reviewer for taking the time to review our manuscript and for the helpful feedback. Specific comments raised by the reviewers are addressed below:

Reviewer Comment 1: Illustrations of the various organoid and organotypic models that have been successfully applied in virology

Reviewer Comment 2: Clear description of which cells were used and whether co-cultured with other cell types or grown alone or as a tissue specimen

Reviewer Comment 3: A table listing the reported cultures with references and advantages - disadvantages 

Our response: The first three comments from the reviewer are focused on being more descriptive about organoid technology. Our aim is to provide a perspective/opinion on organoids for virology rather than a review on organoids. Considering this is a special issue on 3D models, we assume some fore knowledge on the part of the reader. However, as an addition, we chose to refer to existing reviews on organoid technology instead of diluting the message with a complete overview on organoids. Based on reviewers comments, we have now added additional information to section 1 on organoids and their benefits. 

Reviewer Comment 4: In subsection 1 there is an example of HAE organotypic without essential details and no examples of organoids

Our response: A short description on HAE cultures has been added to lines 58-60.

Reviewer Comment 5: Some examples are given later on in the text for MV and Zika viruses. It would be nice to have more information on these studies and add a few more results with other viruses.

Our response: This has been expanded with the inclusion of a section (2.4) on SARS-CoV-2 to provide a topical example.

Reviewer Comment 6: Subsection 3.3 talks about High-throughput screening but it is unclear what the authors want to say. Is this to screen for various viruses or anti-viral drugs or something else? Please give more details.

Our response: We were referring to antiviral screening. This has been clarified on lines 244-245.

Reviewer 2 Report

Sridhar et al provided an opinion on the use of organoids in infectious disease research. While the article is well written and is easy to follow it lacks depth and details in many areas. 

Specific concerns:

  1. Section 1: “organoid technology”. The entire review is based on organoids however the introduction on organoids is very limited. They simply cite another review. This section should be expanded to make it more accessible to people who do not have a large knowledge on the technology.
  2. Line 54 says that they will include HAE under this umbrella. Again HAE should be more defined such as how they are made/isolated, what cells do they contain, why are they included when thinking about organoids etc.
  3. Section 3.2 “Standardization”. I believe that this section could be moved up just under the organoid technology part as it would flow better to have this information just below the introduction to the organoids.
  4. Line 75 the authors state that organoids are used to culture unculturable or hardly culturable viruses such as rotavirus. Rotavirus should be removed from this section, while norovirus needs organoid rotaviruses are grown to high titers in several cell lines.
  5. Line 74-75, this should also be more clarified for norovirus. While organoids can be used to have a first round of infection with norovirus, this virus cannot be propagated efficiently in this model. These are limited one round of infections and organoids are not able to be used to make a “viral stock” that is then used in future experiments. The current version is very misleading.
  6. Section 2.2. “Organoids recapitulate the nature virus host…” The authors state that viruses can be amplified without the need to mutate or adapt. The authors need to clarify this statement more and add examples and references to this whole section.
  7. Line 91-92 – the authors state an example is the dependency on heparin sulfate. However, there are no references and no details about what this is referring to. The authors need to expand and clarify this section.
  8. Line 92-93. The authors state that most cell lines are not polarized and cannot recapitulate polarized organoid cultures. This is not completely true, there are polarized intestinal cells and lung cells. These models work extremely well and have been able to recapitulate most things that are also found in organoids. This statement needs to be softened or expanded to include these models.
  9. Lines 100-105. In this section the authors claim that nectin-4 is the true receptor of MV. How is it written is seems that nectin-4 is only expressed in organoids. Is this true or is it also expressed basolateral in Calu-3 polarized lung cells? If it is also expressed in these cells then there is not proof that organoids are better models for this. This should be clarified in the text.
  10. Section 2.2.2 “Infected organoids resemble…”. This section is lacking many references which should be added.
  11. In general every reference that I checked was incorrect – the authors should take a careful look and make sure to add the correct references to each section.
  12. The authors have a section about Zika virus and clinical implications. It seems very strange given the current pandemic and the number of articles on organoids and SARS-CoV-2 that this was not also highlighted. I would suggest that a full section be added on this topic. Papers such as : Monteil, et al Cell 2020, Lamers et al Science 2020, Stanifer et al Cell Reports 2020, Zang et al Science Immunology 2020 should be used to show how these models are key to evaluate infection.
  13. Lines 130-132. The authors needs to expand and clarify the statement of co-infection of pathogens.
  14. Line 133 – The authors state that host conditions such as age and sex can be used to compare infection. They should list papers and show how this has been used to show differences in infection.
  15. Section “Technical challenges related to biology”. In this section the authors cite their own paper showing that organoids can be made 2D. However, this technique has been used for several years in infectious disease research (Ettayebi, et al Science 2016, Stanifer et al Nature Microbiology, 2020). These papers should be added and highlighted as they actual use these methods to evaluate infection.
  16. Line 155-156. This statement is unclear and should be re-written.
  17. Section “Standardization”. The authors state that the lack of standardization is a major bottleneck. I think that this section should explain this more. Most labs growing organoids are using differentiation factors that are produced from conditioned media which leads to differences between labs. There are also differences in the qualities of additional components that are purchased as well as the Matrigel used to grow the organoids. These issues should be highlighted. While one of the authors works at StemCell, it seems that they would like to push the idea of having a commercially available media for all lab. However, the price of using these medias is out of reach for many labs and most groups will continue to produce their own factors for cost reasons. As these are very real issues faced by groups routinely using organoids for their research, these issues should be highlighted in this section.
  18. Section 4.2 “ Measurement of functionality” This section is very short, it should either be expanded or removed. In its current state is does not bring much to the article.

Author Response

We thank the reviewer for taking the time to review our manuscript and for providing detailed feedback for improvements. We address the specific point raised by the reviewer below:

Response to reviewer 2

Reviewer Comment: Section 1: “organoid technology”. The entire review is based on organoids however the introduction on organoids is very limited. They simply cite another review. This section should be expanded to make it more accessible to people who do not have a large knowledge on the technology.

Our response: Additional information on organoid technology has been added to section 1.

Reviewer Comment: Line 54 says that they will include HAE under this umbrella. Again HAE should be more defined such as how they are made/isolated, what cells do they contain, why are they included when thinking about organoids etc.

Our response: A short description on HAE cultures has been added to lines 58-60.

Reviewer Comment: Section 3.2 “Standardization”. I believe that this section could be moved up just under the organoid technology part as it would flow better to have this information just below the introduction to the organoids.

Our response: We understand the reviewers perspective on this but we feel that this fits better in the technical challenges as this section is focused on perspective/challenges going forward.

Reviewer Comment: Line 75 the authors state that organoids are used to culture unculturable or hardly culturable viruses such as rotavirus. Rotavirus should be removed from this section, while norovirus needs organoid rotaviruses are grown to high titers in several cell lines.

Our response: Rotavirus has been removed from the sentence.

Reviewer Comment: Line 74-75, this should also be more clarified for norovirus. While organoids can be used to have a first round of infection with norovirus, this virus cannot be propagated efficiently in this model. These are limited one round of infections and organoids are not able to be used to make a “viral stock” that is then used in future experiments. The current version is very misleading.

Our response: The following “although it is still limited to one round of infection” has been added to line 85 based on reviewers comment.

Reviewer Comment: Section 2.2. “Organoids recapitulate the nature virus host…” The authors state that viruses can be amplified without the need to mutate or adapt. The authors need to clarify this statement more and add examples and references to this whole section.

Our response: References have been added to section 2.2 to clarify these statements.

Reviewer Comment: Line 91-92 – the authors state an example is the dependency on heparin sulfate. However, there are no references and no details about what this is referring to. The authors need to expand and clarify this section.

Our response: A reference has been added for heparan sulfate adaptation.

Reviewer Comment: Line 92-93. The authors state that most cell lines are not polarized and cannot recapitulate polarized organoid cultures. This is not completely true, there are polarized intestinal cells and lung cells. These models work extremely well and have been able to recapitulate most things that are also found in organoids. This statement needs to be softened or expanded to include these models.

Our response: The statement has been softened to reflect the reviewers comments.

Reviewer Comment: Lines 100-105. In this section the authors claim that nectin-4 is the true receptor of MV. How is it written is seems that nectin-4 is only expressed in organoids. Is this true or is it also expressed basolateral in Calu-3 polarized lung cells? If it is also expressed in these cells then there is not proof that organoids are better models for this. This should be clarified in the text.

Our response: We understand the reviewers confusion. We did not imply that it is only expressed on the basolateral side in the organoid models. However, it is important to point out that the discovery was on HAE and not on polarized calu-3 cells. We have reworded the text on lines 103-104 to avoid misrepresentation.

Reviewer Comment: Section 2.2.2 “Infected organoids resemble…”. This section is lacking many references which should be added.

Our response: More references have been added to this section.

Reviewer Comment: In general every reference that I checked was incorrect – the authors should take a careful look and make sure to add the correct references to each section.

Our response: It appears there was an issue with the referencing software resulting in mismatch between reference numbers. The references have now been fixed.

Reviewer Comment: The authors have a section about Zika virus and clinical implications. It seems very strange given the current pandemic and the number of articles on organoids and SARS-CoV-2 that this was not also highlighted. I would suggest that a full section be added on this topic. Papers such as : Monteil, et al Cell 2020, Lamers et al Science 2020, Stanifer et al Cell Reports 2020, Zang et al Science Immunology 2020 should be used to show how these models are key to evaluate infection.

Our response: We agree with the reviewers comment that the use of organoids for studying SARS-CoV-2 deserves its own section. We have added a full section (2.4) on this topic.

Reviewer Comment: Lines 130-132. The authors needs to expand and clarify the statement of co-infection of pathogens.

Our response: More information has been added to this section to clarify our statements on co-infection along with supporting references.

Reviewer Comment: Section “Technical challenges related to biology”. In this section the authors cite their own paper showing that organoids can be made 2D. However, this technique has been used for several years in infectious disease research (Ettayebi, et al Science 2016, Stanifer et al Nature Microbiology, 2020). These papers should be added and highlighted as they actual use these methods to evaluate infection.

Our response: Additional references suggested by the reviewers have been added.

Reviewer Comment: Line 133 – The authors state that host conditions such as age and sex can be used to compare infection. They should list papers and show how this has been used to show differences in infection.

Our response: References have been added and the sentence has been rephrased.

Reviewer Comment: Line 155-156. This statement is unclear and should be re-written.

Our response: This statement has been expanded to provide clarity.

Reviewer Comment: Section “Standardization”. The authors state that the lack of standardization is a major bottleneck. I think that this section should explain this more. Most labs growing organoids are using differentiation factors that are produced from conditioned media which leads to differences between labs. There are also differences in the qualities of additional components that are purchased as well as the Matrigel used to grow the organoids. These issues should be highlighted. While one of the authors works at StemCell, it seems that they would like to push the idea of having a commercially available media for all lab. However, the price of using these medias is out of reach for many labs and most groups will continue to produce their own factors for cost reasons. As these are very real issues faced by groups routinely using organoids for their research, these issues should be highlighted in this section

Our response: More information has been added to this section. Furthermore, the issue of cost with respect to commercial solutions is made explicit. Conflict of interest regarding author S.S. has been declared in the conflict of interest section.

Reviewer Comment: Section 4.2 “ Measurement of functionality” This section is very short, it should either be expanded or removed. In its current state is does not bring much to the article.

Our response: This section has been expanded to convey our message better.

Round 2

Reviewer 2 Report

I appreciate the efforts made by the authors. The new version of the opinion is much improved.